# On the comparability of frailty scores under the accumulation of deficits approach

**Curtis Huffman**[1]*, **Héctor Nájera**[1], **Mario Ulises Pérez Zepeda**[2]

**1** Programa Universitario de Estudios del Desarrollo, Coordinación de Humandiades, Universidad Nacional Autónoma de México, Ciudad Universitaria, Mexico City, Mexico, **2** Departamento de Investigación, Instituo Nacional de Geriatría, Secretaría de Salud, Mexico City, Mexico

* chuffman@unam.mx

## ⊘ OPEN ACCESS

**Data Availability Statement:** The MHAS (Mexican Health and Aging Study) is partly sponsored by the National Institutes of Health/National Institute on Aging (grant number NIH R01AG018016) in the

## Abstract

### Background

While the cumulative deficit model is arguably the most popular instrument for population-level frailty screening, several questions remain unanswered regarding the comparability of the resulting scores across subpopulations.

### Methods

Based on data from the five waves of the Mexican Health and Aging Study (MHAS) we draw upon the alignment method to test for measurement invariance of frailty scores as per the accumulation of deficits approach.

### Results

Our results show that adjusting for measurement non-invariance not only improves predictive validity of our frailty measures, but resulting scores are more consistent with what is theoretically expected from them in longitudinal research.

### Conclusions

There are clear potential benefits of measurement invariance testing as a general analytical framework from which to tackle with issues of comparability in frailty research.

## Background

In geriatric assessment frailty is generally accepted as a useful concept in understanding the heterogeneity of functional decline observed with chronological aging. It refers to a condition in which the individual is in a vulnerable state at increased risk of adverse health outcomes and/or dying when exposed to a stressor [1].

Without a doubt, identifying frail people (holding a precarious balance between demands and capacity to cope) is of the utmost importance, in particular when it comes to older adults. Not only allows us to better anticipate the burden on healthcare systems, but also may lead to timely interventions that can often have dramatic effects on people's well-being [2]. However,

United States and the Instituto Nacional de Estadística y Geografía (INEGI) in Mexico. Data files and documentation are public use and available at www.MHASweb.org. The authors did not have any special access privileges of any kind.

**Funding:** This work was supported by a research grant from the National Autonomous 385 University of Mexico (DGAPA-UNAM IA300621). The funders had no role in study design, data collection and analysis, decision to publish, or preparation of the manuscript.

**Competing interests:** The authors have declared that no competing interests exist.

**Abbreviations:** AUC, Area under the ROC curve; CD-FI, Cumulative Deficit Frailty Index; IRT, Item Response Theory; MGA, Multiple Group Analysis; MHAS, Mexican Health and Aging Study; ROC, Receiver Operating Characteristic; SEM, Structural Equation Modelling.

it is far from obvious how frailty should be quantified, and the development of measurement tools is still an ongoing process and a research priority.

In population-level screening, the cumulative deficit model elaborated by Rockwood, Mitnitski and colleagues [3] has gained popularity over the years as it is robustly flexible, and the resulting scores strongly associate with mortality and other adverse outcomes with an evident dose-response relationship.

The frailty index relies on the intuition that the more deficits a person has, the more likely that person is to be frail [4]. In accordance, its operationalization involves counting deficits specifically in health (e.g., symptoms, signs, diseases, disabilities or laboratory, radiographic or electrocardiographic abnormalities), and the resulting index is often expressed as a ratio between present deficits divided by the total deficits considered in a given population.

The fact that the cumulative deficit frailty index (CD-FI) can be constructed from data already available in most geriatric assessment surveys, and databases of the sort, is perceived as definite advantage over other alternatives. The fact that different numbers and types of deficits (that fulfill rather modest criteria) may be used in its construction, while preserving a strong association with mortality and other adverse outcomes, makes it robustly flexible and popular among researchers.

However, the fact that the CD-FI is not defined on fixed set of indicators leaves some questions unanswered regarding what we are allowed to infer from differences in frailty scores; that is, its comparability, even within the same sample [5, 6]. This much is widely understood given a CD-FI constructed from a fixed set of deficits, individuals with a frailty score that is relatively high for their age and sex show a significantly increased risk of a range of adverse outcomes; i.e., they are more frail. Hence the importance of studying population norms [7]. But what about, let's say, sex-matched individuals of different age groups sharing the same frailty score? It is far from obvious what we should make of this case, and the proper framing of such questions requires from us to think about the way a given CD-FI (the measurement instrument) interacts with a person's frailty (that which it is set out to measure). This is the general issue of measurement invariance.

A goal in measurement often is making comparisons across population groups using the same index. Ideally, two people with the same frailty score should be in fact equally frail. This type of inference rests on one key assumption: the underlying model and the resulting scores are invariant across the groups of interest. Measurement invariance, therefore, must hold for fair inferences about frailty scores across populations. Otherwise, if a given index measures frailty differently for individuals in different age groups, then we would not be justified in making group comparisons based on that index because it would beg the question: Is the observed difference across groups due to a group difference on how frail they are or due to differences attributable to other sources that are not of interest like measurement error?

It is not easy to overstate the importance of measurement invariance if a researcher wishes to make group comparisons. Meaningful comparisons of statistics, such as means and regression coefficients, can only be made if the measures are comparable across different groups [8]. However, whether a given index interacts with a person's frailty in a comparable way across groups or not, or to what degree, is something that cannot be evaluated in the absence of information regarding the measurement (metrological) model.

Measurement invariance is a testable assumption and it has to do with assessing whether the overall setup of the measurement model is comparable across groups and over time. By measurement model we mean the abstract and idealized (approximate) representation of the interaction between that which we want to measure, in this case the concept of frailty, and the accumulation of health deficits, as instrumental indications provided by the data source (see

[9] for an outline of the model-based approach to the epistemology of measurement). Obviously, the nature of such tests follows that of the measurement model.

Given the operational definition of the CD-FI [4], where the standard procedure for selecting health variables as candidate deficits involves the satisfaction of 4 criteria: 1) being associated with health status, 2) their prevalence must generally increase with age, but 3) must not saturate too early, and 4) cover a range of systems, it stands to reason that the underling measurement model assumes that health deficits are a manifestation of frailty (i.e., a causal or reflective model) and not the other way around (as a formative model would). This kind of measurement model requires that the deficits being considered correlate, as they should if criteria 1 to 3 are satisfied.

It is important to note that it is precisely measurement invariance what is at risk if criterion 3 is not satisfied and the deficits saturate too early. As deficits max out they stop providing us with useful information to make inferences about a person's frailty, ultimately rendering deficits useless for comparisons beyond the point of saturation. But deficits do not need to max out to provide us with *different* information. If a subset of the chosen deficits were to saturate "more quickly" for different population groups or cohorts, comparisons across said groups and time would be hard to interpret. In general terms, if the deficits in health under consideration do not observe somewhat the same modeled relationships between them (correlations in this case) across comparison groups or points in time, chances are that comparisons being made on this basis do not have the meaning we meant for them (i.e., are invalid).

To show the potential of these new practices in frailty research, as an empirical application we use the five waves of MHAS to show the importance of evaluating (and adjusting for) measurement non-invariance in longitudinal research.

## Methods

In all of our estimates we used data gathered by the Mexican Health and Aging Study (MHAS). The MHAS is a national longitudinal study of adults 50 years and older in Mexico. With baseline conducted in 2001, representative of adults born in 1951 or earlier, in 2012 a new sample of adults born between 1952-1962 was added to refresh the sample, and once more in 2018 with adults born between 1963 and 1968.

In order to guarantee reasonable group sizes, we have defined 3-year age-groups in the range of 50 to 88 years of age. Also, to mitigate the risk of survival bias in our estimates, the working sample excludes all those individuals whose age was above 79 (roughly the life expectancy at 50 years of age in Mexico in 2008 [10]) by the time they were detected by the survey for the first time.

In a nutshell, the MHAS (I-V) panel data is shaped into wide form with age-group columns temporarily overlapping the cohort data as illustrated in Table 1.

### CD-FI

For comparability, and following standard procedures [4], we constructed a 35-deficit CD-FI based on [11, 12]. Deficits included functional status, chronic diseases, self-rated health, cognitive status, and depressive symptoms. All self-reported. Further details about selection, coding and screening (syntax included) can be found in [11].

### Health outcomes

Mortality was recovered from next-of-kin data. MHAS assess falls by asking "Have you fallen down in the last two years?", whose positive answer elicits the question "Approximately how many times has this happened?". Fall syndrome was defined as having answered >2 to this last

**Table 1. Number of observations by age groups and cohorts in the MHAS-ALD.**

| Age groups/ Cohort | Average age (sd) | | | | | | | | | | | | | *n* |
|---|---|---|---|---|---|---|---|---|---|---|---|---|---|---|
| | 51 (0.82) [50–52] | 54 (0.81) [53–55] | 57 (0.82) [56–58] | 60 (0.82) [59–61] | 63 (0.81) [62–64] | 66 (0.82) [65–67] | 69 (0.82) [68–70] | 72 (0.82) [71–73] | 75 (0.82) [74–76] | 78 (0.82) [77–79] | 81(82) [80–82] | 84(81) [83–85] | 87(81) [86–88] | |
| 15 | 1,942 | - | - | - | - | - | - | - | - | - | - | - | - | 1,942 |
| 14 | 483 | 1,863 | - | - | - | - | - | - | - | - | - | - | - | 2,346 |
| 13 | 1,314 | 1,317 | 1,469 | - | - | - | - | - | - | - | - | - | - | 4,100 |
| 12 | - | 1,472 | 1,417 | 1,344 | - | - | - | - | - | - | - | - | - | 4,233 |
| 11 | - | - | 1,502 | 1,460 | 1,353 | - | - | - | - | - | - | - | - | 4,315 |
| 10 | 663 | - | - | 1,400 | 1,340 | 1,213 | - | - | - | - | - | - | - | 4,616 |
| 9 | 2,116 | 1,951 | - | - | 1,853 | 1,804 | 1,560 | - | - | - | - | - | - | 9,284 |
| 8 | - | 1,968 | 1,837 | - | - | 1,618 | 1,576 | 1,354 | - | - | - | - | - | 8,353 |
| 7 | - | - | 1,724 | 1,599 | - | - | 1,384 | 1,317 | 1,118 | - | - | - | - | 7,142 |
| 6 | - | - | - | 1,487 | 1,376 | - | - | 1,086 | 997 | 842 | - | - | - | 5,788 |
| 5 | - | - | - | - | 1,286 | 1,188 | - | - | 921 | 810 | 641 | - | - | 4,846 |
| 4 | - | - | - | - | - | 1,201 | 1,109 | - | - | 759 | 658 | 513 | - | 4,240 |
| 3 | - | - | - | - | - | - | 990 | 907 | - | - | 558 | 453 | 304 | 3,212 |
| 2 | - | - | - | - | - | - | - | 744 | 679 | - | - | 349 | 244 | 2,016 |
| 1 | - | - | - | - | - | - | - | - | 672 | 583 | - | - | 258 | 1,513 |
| Total | 6,518 | 8,571 | 7,949 | 7,290 | 7,208 | 7,024 | 6,619 | 5,408 | 4,387 | 2,994 | 1,857 | 1,315 | 806 | 67,946 |

Note: Cohort 1 refers to all those born between 1925 and 1927, seen by the MHAS sometime between 2001 and 2018 with ages 74–88; Cohort 2 [1928-1930], ages 71–88; Cohort 3 [1931-1933], ages 68–88; Cohort 4 [1934-1936], ages 65–85; Cohort 5 [1937-1939], ages 62–82; Cohort 6 [1940-1942], ages 59–79; Cohort 7 [1943-1945], ages 56–76; Cohort 8 [1946-1948], ages 53–73; Cohort 9 [1949-1951], ages 50–70; Cohort 10 [1952-1954], ages 50–67, Cohort 11 [1955-1957], ages 56–64, Cohort 12 [1958-1960], ages 53–61; Cohort 13 [1961-1963], ages 50–58; Cohort 14 [1964-1966], ages 50–55 and Cohort 15 [1967-1969], ages 50–52.

Source: Prepared by the authors based on data from MHAS I–V

question. Gait speed and handgrip strength tests were only taken for a subsample of individuals in 2012. Gait speed refers to the time (best out of two tries) it takes an individual to cross the first foot over the end of a 4-meter strip, where a threshold of 0.8 *m/s* was used to define abnormality. For handgrip tests, cut-off values of 20 and 30 kilograms were used for women and men, respectively.

## Alignment method

Invariance analysis is a method that assesses whether the underlying measurement model of a scale is equivalent across the groups of interest. To do so this technique looks at the similarity of parameters of the underlying measurement model given that these constitute the blueprint of the observed scores. For the CD-FI there are two key parameters of interest: factor loadings and thresholds. The first set tell the amount of explained variance of a given item. Ideally, this amount should be equivalent across groups otherwise the observed scores will reflect these unwanted differences. Thresholds refer to the level of frailty at which one person experiences a given deficit. In principle, two people with the same levels of frailty should be equally likely to present the observed deficit.

It is important to note that, under measurement invariance, we do not expect from the CD-FI's different components to exhibit the same prevalence across comparison groups. Rather, what we do expect is that the amount of the frailty score's variability explained by each component should be roughly the same. In other words, what is expected is that, for a given

level of frailty, the likelihood of expressing a particular health deficit is roughly the same, irrespective of the population subgroup to which individuals belong.

Needless to say, measurement invariance is a regulative ideal that can only be approximated in practice [13], which makes its quantification all the more important. There are two main different methods to look at the equivalence of the parameters across groups: Multiple Group Analysis (MGA) and the Alignment Method. Both belong to the family of techniques within the field of structural equation modelling (SEM).

Traditionally, assessing measurement invariance is conducted under the MGA approach by way of Confirmatory Factor Analyses, which requires some statistical expertise in Structural Equation Modelling, and become problematic when many groups are tested. In contrast, the Alignment Method [14], as a (somewhat) recently developed alternative, approaches measurement invariance as an optimization problem. An innovation that fully automates the procedure of identifying noninvariant deficits (items) while estimating a model that allows for group mean comparisons.

In this paper we resort to the Alignment Method to obtain a CD-FI model that allows for age-group comparisons via latent scores. While it can also be used to improve particular CD-FIs by screening for invariant deficits so that (observed) raw scores can be used.

Unlike traditional MGA approaches to measurement invariance, the Alignment Method is not a measurement invariance testing procedure in itself, which makes it more suitable when dealing with a large number of groups. It is rather a treatment of measurement invariance as an optimization problem [15]. Its goal is to estimate the simplest model with the largest amount of invariance —that is, a model with factor loadings and intercepts/thresholds (item-level parameters) that are as close to equivalent as possible. As long as researchers are dealing with minor measurement differences (approximate measurement invariance), the alignment method produces a factor model that is sufficient to make (unbiased) factor mean comparisons by selecting latent scores means and variances that minimize measurement non-invariance of the item-level parameters. Of course, if the assumption of approximate measurement invariance is violated the simplest and most invariant model may not be the true model [16].

While the alignment optimization was not designed to evaluate whether instruments are approximate invariant (as this is an assumption of the optimization procedure), and does not allow for the testing of specific models with differing levels of measurement invariance, if the methods assumptions hold, the results indicate which items are non-invariant. Unfortunately, there is no existing package in R that replicates the alignment method, and all of our estimates were obtained using *Mplus* 8.5. A brief tutorial on the alignment method is provided in [17].

## Predictive performance

In the absence of a reference method to measure frailty, to further strengthen our confidence in the added value of adjusting for measurement non-invariance in CD-FI construction, we follow the common practice of comparing its relative performance in predicting adverse health outcomes (predictive validity) vis-à-vis the traditional (raw) version of the CD-FI and its centiles (according to the individual's sex and age-group). Particularly, we used logistic regressions for mortality risk, fall syndrome, low gait speeds ($<8$ $m/s$) and handgrip strength ($<20$ for women and $<30$ kilograms for men). All regressions included both age and age squared with a full set of sex interactions. For comparability purposes regarding the relative importance of the frailty scores, and help with the interpretation, x-standardized coefficients (given the frailty scores have different scales) are reported along with a measure of area under the receiver operating characteristic (ROC) curve –a graph of sensitivity versus one minus specificity as the cutoff $c$ is varied. A model with no predictive power has a ROC area of 0.5 while a perfect model has an area of 1.

## Results

### Alignment

A 13-group alignment analysis of the 35 items is performed for the 13 time points by sex separately. The sex-specific analyses are meant to keep the focus on cross-time comparisons as CD-FI dynamics are found to be strongly sex-sensitive [18]. The results of the 13-group analyses are shown in Tables 2 and 3. The tables indicate which item parameters, thresholds and loadings, are non-invariant in which groups with asterisks and plus signs, respectively. It is seen that, for both women and men, there are more non-invariant thresholds (24% and 16%) than loadings (5% and 3%). Well under the 25% rule of thumb for trustworthy alignment results mentioned in [19] and supported by simulations in [20].

The results in Tables 2 and 3 also include the alignment R-squared measure, which is meant to be interpreted as the proportion of variation across groups of the parameters, intercepts and loadings, explained by variation in the factor mean and variance [respectively] across groups [21]. In this way, values close to 1 are associated with invariant parameters, while values closer to 0 are generally associated with non-invariant parameters.

For developing and evaluating a particular CD-FI to modify or improve it by ensuring there is invariance, it is of interest to see which deficits and which age-groups contribute most to the non-invariance. It is found that among the most invariant deficits are self-rated health (*health*), help with finances (*fin*), loss of appetite in the last 2 years (anorexia). [11] had already identified *health* as one the most relevant deficits based on a Markov network analysis.

Cancer is among the deficits that contribute the least to the non-invariance of the CD-FI, even though the associated parameters show no significant difference across age-groups for women. This result largely agrees with [12], who find the cancer diagnosis as marginally independent from other deficits. The fact that it does not show asterisks or plus signs is due to large errors given its low prevalence, particularly around the early 70s, resulting in statistical tests of low power and the highest severity under the model (takes the longest to activate) as well as providing the least information to discriminate between levels of frailty.

Next to *cancer*, *bed* seems to contribute the least to the CD-FI's comparability, but for different reasons. By all appearances, spend one day or more in bed due to sickness or injury is associated with higher levels of frailty before reaching 70 years of age than after. While, at the same time, its activation takes a relatively long time to build up (low discrimination) rendering of little use in distinguishing but the largest differences in frailty.

Perhaps unsurprisingly, not doing exercise or hard physical work at least 3 times a week (*exercise*) seems to mean quite different things through the life course in terms of frailty. For younger women it is even likelier to be associated with more vulnerable health states. This makes sense if, for example, in contexts of low preventive care access, women younger than 50 years of age hardly ever do physical work 3 times a week unless prescribed by a doctor at the onset of a serious illness. Be that as it may, the deficit *exercise* seems to "activate" (going from 0 to 1) at different levels of frailty for different age-groups (see the asterisks at the bottom of Table 2), thus providing different information and compromising the interpretation of the frailty score.

Something similar happens to deficits related to depressive symptoms (*effort*, *depress*, *happy*), they seem to be related with higher levels of frailty before reaching 70 years of age.

Table 3 shows noticeable contrasts with Table 2 that warrant further inspection. First, Table 3 shows more invariance in general (see the instrumental activities of daily living: *shop*, *meals*, *meds*, *fin*).

However, there are also important similarities. Exercise exhibits the same pattern of high errors in the first age-groups, lower discrimination, but decreasing severity (difficulty, easier to

**Table 2. Invariance results for aligned thresholds (\*) and loadings (+) parameters for all deficits considered (Women).**

| Deficits | $R^2_*$ | $R^2_+$ | Age groups | | | | | | | | | | | | |
|---|---|---|---|---|---|---|---|---|---|---|---|---|---|---|---|
| | | | 50–52 | 53–55 | 56–58 | 59–61 | 62–64 | 65–67 | 68–70 | 71–73 | 74–76 | 77–79 | 80–82 | 83–85 | 86–88 |
| dress | 0.992 | 0 | | | | | | | | | | | | | |
| chair | 0.883 | 0 | + | + | | | | | | | | | | | |
| walk | 0.744 | 0.328 | | | | | | | | | | | | | |
| eat | 0.694 | 0.121 | | | | | | * + | | | | | | | * |
| groom | 0.582 | 0.245 | | | | | | + | | | | * | * + | * | * |
| toilet | 0.734 | 0 | | | | | | | | | | | | | * |
| stairs | 0.913 | 0 | + | + | + | | | | | | | * | * | * | |
| lift | 0.904 | 0 | | | + | + | | | * | * | * | * | * | * | * |
| shop | 0.821 | 0.447 | | | | | | | * | * | * | * | * | * | |
| meals | 0.834 | 0.135 | | | | | | | | | | * | * | * | * |
| meds | 0.833 | 0.651 | | | | | | | | | | * | * | * | * |
| fin | 0.881 | 0.842 | | | | | | | | | * | | * | | |
| weight | 0.737 | 0.341 | | | | | | | * | | * | * | * | * | * |
| health | 0.757 | 0.85 | | | | | | | | | | | * | | |
| change | 0.536 | 0.351 | | | | | | | | * | * | | | * | |
| bed | 0 | 0 | * | * | * | * | * | * | | | | | + | | |
| tired | 0 | 0.563 | | | | * | * | * | * | * | * | * | * | * | * |
| w_out | 0.952 | 0 | | | | | | | | | | * | * | | * |
| effort | 0 | 0.53 | * | * | * | * | * | | | | * + | + | * | + | + |
| depres | 0 | 0.542 | * | * | * | * | * | * | * | * | | + | | | |
| happy | 0 | 0.579 | * | * | * | * | * | * | * | * | | | | | |
| lone | 0.386 | 0.532 | * | * | * | * | | | | | * | | | | |
| noener | 0.731 | 0 | * | | | | | | | | | | | | |
| hyper | 0.722 | 0.333 | * | * | * | | | | | | | | | | + |
| heart | 0.92 | 0.296 | | | | | | | | | | | | | |
| chf | 0.587 | 0.055 | | | | | | | | | | | | | |
| stroke | 0.824 | 0 | | | | | | | | | | | | | |
| cancer | 0 | 0 | | | | | | | | | | | | | |
| diabet | 0 | 0.493 | * | | | | | | | | | * | * | * | * |
| arth | 0.819 | 0.551 | | | | | | | | | | | | | |
| cld | 0.905 | 0.483 | | | | | | | | | | | | | |
| memory | 0.922 | 0.482 | | | | | | | | | | | | | |
| grip | 0.993 | 0.492 | | | | | | | | | | | | | |
| anorex | 0.93 | 0.622 | | | | | | | | | | | | | |
| exer | 0.455 | 0 | * + | * + | * + | * + | * | * | * | | | | | * | * |

Note: Asterisks (plus signs) refer to deficits that show significant non-invariance for the threshold (loading) parameter.

Source: Prepared by the authors based on data from MHAS I–V

activate as they grow old, less frailty is needed for its activation). It is important to note that the opposite seems to be the case with women, for whom the activation of this deficit seems to be delayed as they grow older. Indeed, *exercise* seems to mean different things in terms of frailty by sex and age-group. Also *bed* remains among the most non-invariant deficits for men, as self-rated health (*health*) and loss of appetite in the last 2 years (*anorexia*) remain among the deficits that contribute the most to the CD-FI comparability.

**Table 3. Invariance results for aligned thresholds (\*) and loadings (+) parameters for all deficits considered (Men).**

| Deficits | $R^2_*$ | $R^2_+$ | Age groups | | | | | | | | | | | | |
|---|---|---|---|---|---|---|---|---|---|---|---|---|---|---|---|
| | | | 50–52 | 53–55 | 56–58 | 59–61 | 62–64 | 65–67 | 68–70 | 71–73 | 74–76 | 77–79 | 80–82 | 83–85 | 86–88 |
| dress | 0.961 | 0.492 | | | | | | | | | | | | | |
| chair | 0.947 | 0 | | | | | | | | | | | | | |
| walk | 0.395 | 0.08 | * + | * | | | | | | | | | | | |
| eat | 0.673 | 0.29 | | | | | | | | | | | | | |
| groom | 0.576 | 0.272 | | | | | | | | | | | | | |
| toilet | 0 | 0 | * | | | | | | | | | | | | |
| stairs | 0.978 | 0 | | | | | | | | | * | | | | |
| lift | 0.976 | 0 | | | | | | | | | | * | * | * | |
| shop | 0.666 | 0 | + | + | | | | | | | | | | | |
| meals | 0.656 | 0 | + | + | + | + | + | | | | | | | | |
| meds | 0.824 | 0.041 | | | | | | | | | | | | | |
| fin | 0.886 | 0 | | | | | | | | | | | | | |
| weight | 0.913 | 0.618 | | | | | | | | | | | | | |
| health | 0.893 | 0.709 | | | | | | | | | | | * | * | * |
| change | 0.784 | 0.883 | | | | | | | | | | | | * | * |
| bed | 0 | 0 | | | | | | | * | * | * | * | * | * | * |
| tired | 0 | 0.6 | * | * | * | * | | | | | * | * | * | * | * |
| w_out | 0.961 | 0 | | | | | | | | | | | | | |
| effort | 0.015 | 0.641 | * | * | * | * | | | | | | * | * | * | * |
| depres | 0.059 | 0.609 | * | * | * | * | | | | | | | * | * | * |
| happy | 0 | 0.512 | | | | | | * | * | * | * | * | * | * | * |
| lone | 0.936 | 0.672 | | | | | | | | | | | | | |
| noener | 0.856 | 0 | | | | | | | | | | | | | |
| hyper | 0.827 | 0.465 | | | | | | | | | | | | | |
| heart | 0.819 | 0.326 | | | | | | | | | | + | | | |
| chf | 0.86 | 0.305 | | | | | | | | | | + | | | |
| stroke | 0.896 | 0 | | | | | | | | | | | | | |
| cancer | 0.49 | 0.12 | | | | | | | | | * | * | * | * | * |
| diabet | 0 | 0.162 | | | | | | | | | * | | * | * | * |
| arth | 0.844 | 0.488 | | | | | | | * | | | | | | |
| cld | 0.906 | 0.359 | | | | | | | | | | | | | |
| memory | 0.888 | 0.56 | | | | | | | * | | * | | | | |
| grip | 0.956 | 0.72 | | | | | | | | | | | | | |
| anorex | 0.903 | 0.506 | | | | | | | | | | | | | |
| exer | 0.599 | 0 | * + | * + | * + | * + | * + | * | * | * | | | | | |

Note: Asterisks (plus signs) refer to deficits that show significant non-invariance for the threshold (loading) parameter.

Source: Prepared by the authors based on data from MHAS I–V

A subproduct of the alignment optimization method are the frailty scores that result from the factor model with the largest amount of invariance; that is, the frailty scores that are as comparable as possible. Figs 1 and 2 show the distribution of the resulting frailty scores, raw and adjusted for measurement non-invariance, respectively.

Two things are worth noting in looking at these figures. First of all, while in both cases mean frailty scores grow as age groups grow older, unlike the raw scores, the aligned frailty scores also reduce their dispersion. This measure of convergence is exactly what one would

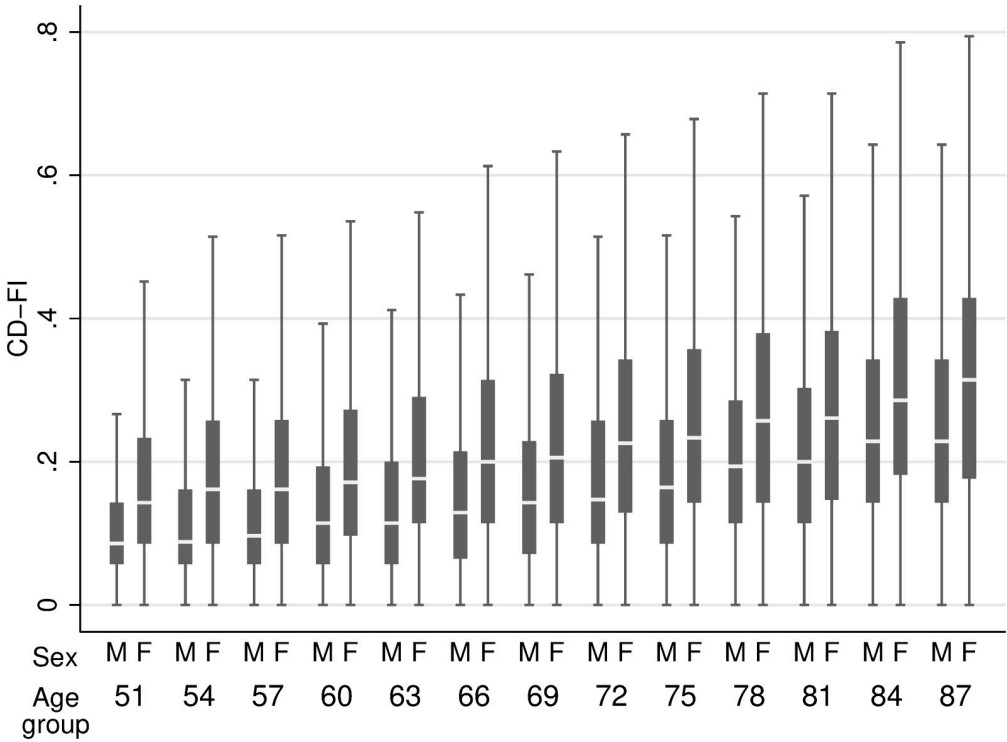

**Fig 1. Raw frailty scores distribution by sex and age group.**

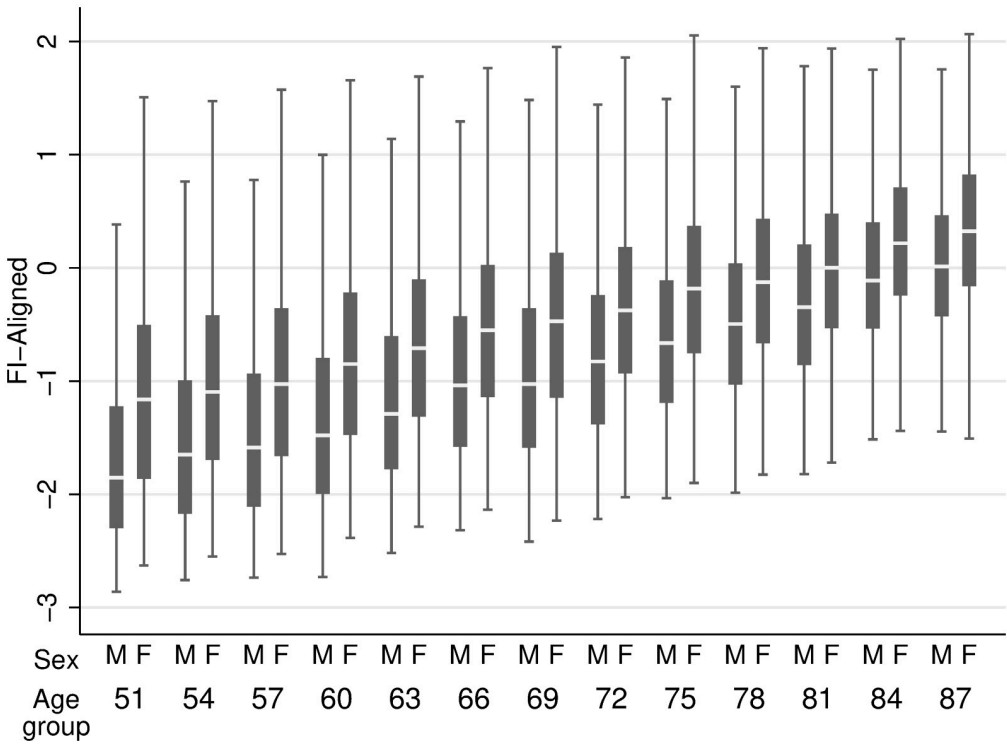

**Fig 2. Aligned frailty scores distribution by sex and age group.**

Table 4. Logistic regression results for adverse outcomes risks (x-standardized coefficients).

| | | Raw | Centiles* | Aligned |
|---|---|---|---|---|
| Mortality | Log-odds | 0.661 | 0.722 | 0.907 |
| (n = 67,946) | AUC[†] | 0.789 | 0.777 | 0.796 |
| Fall syndrome | Log-odds | 0.574 | 0.587 | 0.7166 |
| (n = 67,946) | AUC[†] | 0.697 | 0.696 | 0.7023 |
| Gait speed | Log-odds | 0.670 | 0.565 | 0.724 |
| (n = 3,325) | AUC[†] | 0.755 | 0.750 | 0.758 |
| Handgrip strength | Log-odds | 0.409 | 0.342 | 0.469 |
| (n = 3,343) | AUC[†] | 0.681 | 0.679 | 0.684 |

*Centiles by sex and age group

[†] Area under ROC curve

expect of a successful frailty measure: An age at which virtually all people are frail [22]. Second, the aligned frailty scores also exhibit a measure of convergence between sex. Again, the divergence exhibited by the raw scores between men and women as age groups grow older is hard to accommodate with what we would expect from a successful frailty measure.

## Predictive performance

Table 4 shows the standardized coefficient (the log odds times the standard deviation of the corresponding frailty score) of the logistic regressions to assess the effect of adjusting the CD-FI for measurement non-invariance. There we can see that, in all four cases (panels), the adjusted CD-FI (Aligned, fifth column) exhibits greater predictive performance than the traditional and centilized approaches (Raw and Centiles, third and forth columns, respectively). We see in the first panel, for example, that a 1 standard deviation increase in the aligned CD-FI produces, on average, a 0.907 increase in the log odds of mortality within three years.

Note that, if only marginally, also the area under the ROC curve is larger for the aligned CD-FI in all four cases.

## Discussion

Quantifying frailty requires an abstract representation of the measurement process; i.e., a measurement (metrological) model [23]. That this model holds (is equally applicable) in each group under contrast is a fundamental assumption underlying every conclusion derived from such comparisons. Whether there are good reasons behind this assumption or not is not only a theoretical, but also an empirical issue. In the absence of an impeccable measure of frailty (a so-called "gold standard"), the burden of the proof falls with those making conclusions based on the comparability of the scores.

As a great deal of theoretical and empirical research on measurement invariance has been conducted within the contexts of Factor Analysis and Item Response Theory (IRT), particularly as the issue pertains to psychological and educational assessment [24], it naturally begs the question of their pertinence in the case of the CD-FI.

Our results in the previous section are predicated on the acceptance that, in the measurement model underlying the CD-FI, frailty exist at a deeper conceptual level than the health deficits, and that the latter are consequences (effects) the former. Not everybody agrees [25, 26]. On this matter, [27] suggests a mental experiment to help researchers to think about their measurement models: Imagine a change in a person's frailty, net of other health deficit influences.

Will this lead to a change in deficits? If the answer is on the negative, you will hardly find much value in our results. Note also that the answer to this question also has implications for what we should expect in terms of frailty from specific deficit repair, as well as the informational value of data-driven frailty profiles [28].

However, it is important to note that if, as Drubbel et al. [25] and Xue & Varadhan [26] argue, the measurement model underling the CD-FI does not consider health deficits as effects of frailty (but the syndrome itself), and consequently there is no good reason why deficits should correlate in the first place, while this would invalidate our results, this does not exempt researchers from providing evidence in favor of the comparability of the frailty scores in terms of the corresponding measurement model, it would simply shift the focus of invariance testing away from the observed correlations between deficits. In other words, however modeled the measurement process, measurement invariance is an issue that every group comparison needs to address.

On the other hand, if our assumption of the measurement model underlying the CD-FI being reflective makes sense, our results may prove important for frailty measurement, and our understanding of the heterogeneity of aging population, as measurement invariance testing provides a robust framework (and new avenues) to investigate frailty differences across population groups.

Underpinning our confidence in contrasts between people of different sex and age-group has the potential of furthering our understanding of subclinical frailty (or pre-frail, as it is usually calculated based on the whole distribution of frailty scores in a population [29]), pathways that underlie frailty (individual longitudinal trajectories), and even the well-known male-female health-survival paradox [30].

As measurement invariance is a fundamental assumption in making comparisons between population groups, as well as between individuals, it is also important to note that, even though our focus in this article has been on population-level screening based on secondary data analysis, the alignment method may also prove useful for the design of frailty measures fitted to guide clinical care decisions –by way of assessing which items behave differently across groups of individuals, provided a reflective measurement model make sense in the first place.

Admittedly, our age-group invariance analysis may be confounded by cohort effects. As shown in Fig 1, any given age-group in our data comes from different cohorts up to (roughly) 30 years apart, and any given cohort only takes us as far as 18 years (the distance in time between MHAS' waves V and I) in the best of cases. It may well be that our invariance results are more a result of a cohort effect rather than an age-group effect. Fortunately, the alignment method can help us to untangle this whenever the sample size allows for it. Notice, for example, that we only have 258 individuals born between 1925 and 1927 in the 86–89 age-group, which would render the alignment results somewhat unreliable for this specific group. Nevertheless, whatever the source of non-invariance, comparability (and inference) requires its minimization.

## Conclusion

In this paper we focus on practical considerations surrounding the comparability (same meaning) of the CD-FI across subpopulations. While the nature of the relationship between the CD-FI, as an instrumental indication, and the concept of frailty (an individual's vulnerability to adverse health outcomes) remains somewhat undertheorized and, as a consequence, the measurement model underling the CD-FI is still contested, there is hardly a better analytical framework than measurement invariance to tackle these general issues. We believe our results show the potential benefits of this approach for frailty measurement development, particularly of the alignment optimization method whenever multi-group factor analysis proves pertinent.

Without a doubt, more work is necessary to reap the full benefits of measurement invariance testing in advancing our understanding of frailty measurement, but we believe pursuing such work will inevitably add greater clarity also into its mechanisms and management.

## Author Contributions

**Conceptualization:** Curtis Huffman.

**Data curation:** Mario Ulises Pérez Zepeda.

**Methodology:** Curtis Huffman, Héctor Nájera.

**Validation:** Héctor Nájera, Mario Ulises Pérez Zepeda.

**Writing – original draft:** Curtis Huffman.

**Writing – review & editing:** Héctor Nájera, Mario Ulises Pérez Zepeda.

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
