## [Decision Letter · Decision Letter 0]

13 Jul 2023

PONE-D-23-10946On the comparability of frailty scores under the accumulation of deficits approachPLOS ONE

Dear Dr. Huffman,

Thank you for submitting your manuscript to PLOS ONE. After careful consideration, we feel that it has merit but does not fully meet PLOS ONE’s publication criteria as it currently stands. Therefore, we invite you to submit a revised version of the manuscript that addresses the points raised during the review process.

We look forward to receiving your revised manuscript.

Kind regards,

Guillermo Salinas-Escudero, PhD. MsC.

Academic Editor

PLOS ONE

Journal Requirements:

“This work was supported by a research grant from the National Autonomous

385 University of Mexico (DGAPA-UNAM IA300621)”

Additional Editor Comments:

Dear authors,

Regarding the article “On the comparability of frailty scores under the accumulation of deficits approach” submitted by you to Frontiers... I want to inform you that it has been evaluated by two reviewers with opposite responses. I tried to have the article reviewed by a third expert, but all refused to review the manuscript. Consequently, I decided to continue with the evaluation process. I request that each reviewer’s comments be considered to achieve a sufficient quality manuscript to be published.

Reviewers' comments:

Reviewer's Responses to Questions

**Comments to the Author**

1. Is the manuscript technically sound, and do the data support the conclusions?

Reviewer #1: Yes

Reviewer #2: Yes

2. Has the statistical analysis been performed appropriately and rigorously? 

Reviewer #1: Yes

Reviewer #2: Yes

3. Have the authors made all data underlying the findings in their manuscript fully available?

Reviewer #1: Yes

Reviewer #2: Yes

4. Is the manuscript presented in an intelligible fashion and written in standard English?

Reviewer #1: Yes

Reviewer #2: Yes

5. Review Comments to the Author

Reviewer #1: The authors have applied a thoughtful approach to investigating certain properties (measurement invariance) of the frailty index. The manuscript is written well and all statistical analyses appear sound. This piece of work was enjoyable to read and I have no reservations about its publication.

Reviewer #2: Review: PONE-D-23-10946

General comments:

The authors use data from the large Mexican Health and Aging Study to examine the performance of a 35-component deficit accumulation frailty index (FI) across age groups and time. They find compelling evidence that their FI is not invariant across groups, i.e., that the explained variance and the thresholds (i.e., association between the prevalence of an individual component and the FI score) are heterogeneous. They use an analytical approach (alignment method) to reduce invariance among groups and demonstrate that it provides a more consistent description of sex-related differences across age. They also demonstrate impressive expertise in scaling methods.

The authors tackle a difficult task in that, as they aptly note, the concept of frailty is “somewhat undertheorized.” I find it unsurprising that their FI is non-invariant across age groups. Attempts seen in biomedical literature to use different FIs, based on different modes of assessments, to compare groups between studies and to set general cutpoints for defining “frailty” seem to me to be fraught with issues.

The accumulation of deficits, i.e., which deficits occur for an individual and when, varies depending on stressors, genetics, etc. The rates that these occur likely vary with biological and chronological aging and with individuals’ reserves of resilience. Doesn’t this “naturally” induce some level of non-invariance?

As noted above, the authors propose the alignment method as a means to enhance group comparisons, i.e., allowing the weights of individual components to vary among groups, which appears to be akin to using modestly different FIs. This approach has empirical-based value, but is not likely to have value for what I feel are the most important uses of FIs, i.e., to inform and help guide clinical care decisions and to identify approaches to slow increases in FI over time. Towards these goals, wouldn’t it be preferable to have a more well-defined and less malleable FI?

Lacking is a discussion of the use of FIs as outcomes in clinical trials, in which comparison groups are based on randomization. Is non-invariance of less importance in such studies?

The authors bring impressive analytical skills to this research project. Their findings are limited to a single FI – there are many, many in the literature. However, I think it is to be expected that similar findings would generalize to other FIs.

I think the greatest value of this manuscript is that it sheds light on the natural limitations of FI comparisons across groups and studies. I also think that this should be understood as an inherent limitation of these metrics. In the end, however, I don’t expect that the recommended alignment method will be widely adopted for FIs. I expect that, with care, FIs will continue to have valid uses in informing clinical practice and in clinical trials.

Specific comments:

Lines 1-6: Very well put.

Line 80: The expectation that components should be correlated – I’m not sure I follow this (and what seems to be a counter example, cancer, is discussed later). Might relationships with frailty not be orthogonal?

Line 84: “max out” – certainly such components do not help with future rates of accumulation, but if comparisons are with less frail subgroups, are these still not useful?

Line 106: “minimize” – perhaps a different word choice, in that it is not clear that minimization has occurred.

Line 118: “self-reported” – a limitation, correct? Is this likely to contribute to some non-invariance?

Section 2.3: It is not clear to me why the amount of “explained variability” should be equal for a given component across subgroups. Isn’t it to be expected that the prevalence of components should vary depending on biological age so that some relationships may be more attenuated or vary depending on one’s underlying health status?

6. PLOS authors have the option to publish the peer review history of their article (what does this mean?). If published, this will include your full peer review and any attached files.

Reviewer #1: No

Reviewer #2: No

---

## [Author Response · Author response to Decision Letter 0]

23 Aug 2023

Thank you for the kind invitation to resubmit our work (PONE-D-23-10946). You’ll find our detailed answers to each comment below. 

Comments for the Author:

Reviewer #1: 

1. The authors have applied a thoughtful approach to investigating certain properties (measurement invariance) of the frailty index. The manuscript is written well and all statistical analyses appear sound. This piece of work was enjoyable to read and I have no reservations about its publication.

We appreciate the comment and express our gratitude for taking the time of reviewing our work.

Reviewer #2: 

1. The authors use data from the large Mexican Health and Aging Study to examine the performance of a 35-component deficit accumulation frailty index (FI) across age groups and time. They find compelling evidence that their FI is not invariant across groups, i.e., that the explained variance and the thresholds (i.e., association between the prevalence of an individual component and the FI score) are heterogeneous. They use an analytical approach (alignment method) to reduce invariance among groups and demonstrate that it provides a more consistent description of sex-related differences across age. They also demonstrate impressive expertise in scaling methods.

We appreciate the comment and express our gratitude for taking the time of reviewing our work.

2. The authors tackle a difficult task in that, as they aptly note, the concept of frailty is “somewhat undertheorized.” I find it unsurprising that their FI is non-invariant across age groups. Attempts seen in biomedical literature to use different FIs, based on different modes of assessments, to compare groups between studies and to set general cutpoints for defining “frailty” seem to me to be fraught with issues.

Needless to say we agree on this observation. We see our paper as contributing with some methodological clarity as to the nature of these issues. It is our hope that, by taking our methodological approach, researchers will be better equipped to work around some of the problems surrounding FIs’ comparability.

3. The accumulation of deficits, i.e., which deficits occur for an individual and when, varies depending on stressors, genetics, etc. The rates that these occur likely vary with biological and chronological aging and with individuals’ reserves of resilience. Doesn’t this “naturally” induce some level of non-invariance?

It is indeed to be expected that the accumulation of deficits varies with biological and chronological aging. Among the things we know for sure about frailty is that it grows with age. It is precisely this correlation what is being leveraged by frailty indices (FI) to indicate different levels of reserves of resilience. Indeed, the concepts of frailty and biological-age are not that far apart. In this sense, we do not see this as a threat to FIs comparability, but the very thing we want to “distill”. As figure of speech, we could say that people with the same frailty score have the same biological-age irrespective of their chronological ages. The threats to comparability start when deficits relate differently to each other (presumably as a function of frailty) across comparison groups (population groups or cohorts) , but are treated equally in scoring individuals. This is what we believe is important to keep track of in terms of comparability of scores, and we propose to include the alignment method as part of researchers’ toolbox to start doing so. 

4. As noted above, the authors propose the alignment method as a means to enhance group comparisons, i.e., allowing the weights of individual components to vary among groups, which appears to be akin to using modestly different FIs. This approach has empirical-based value, but is not likely to have value for what I feel are the most important uses of FIs, i.e., to inform and help guide clinical care decisions and to identify approaches to slow increases in FI over time. Towards these goals, wouldn’t it be preferable to have a more well-defined and less malleable FI?

Yes, of course. While a subproduct of the alignment method are scores as comparable as possible, which can enhance FI comparability in research settings., they are not a useful clinical tool. Measurement non-invariance can be particularly damaging in secondary data analysis settings, and it is in this settings where we see the most immediate and valuable application of our approach. However, the alignment method can also be used by FI developers to reach more well-defined and indeed less malleable FIs, by iteratively dropping the most invariant items (deficits), for example. We are clearer on this matter in the new version. 

5. Lacking is a discussion of the use of FIs as outcomes in clinical trials, in which comparison groups are based on randomization. Is non-invariance of less importance in such studies?

This is an interesting point as measurement models are indeed causal models, and clinical trials are particularly suited to make causal inference; that is, to attribute causality to a particular treatment. The main focus of our paper, however, is not about treatment effect inference, but how to measure an outcome that could inform those treatment effect assessments. In this sense, a well-developed clinical trial may disperse every doubt regarding sample selection/bias, and still not get us any closer to a proper measure to assess the treatment effect. 

Alas, an experiment that could give us information regarding how to better measure frailty would require a frailty measure, as we would have to induce it to subjects to gauge the change in our instruments. For the moment being all we have is the theoretical soundness of our measurement model. 

6. The authors bring impressive analytical skills to this research project. Their findings are limited to a single FI – there are many, many in the literature. However, I think it is to be expected that similar findings would generalize to other FIs.

This indeed is an empirical question we would like to see explored in future studies. We believe it to be an interesting question. Our hypothesis is that we will be able to discard certain deficits from the list of candidates that today pass first inspection in FI construction. 

7. I think the greatest value of this manuscript is that it sheds light on the natural limitations of FI comparisons across groups and studies. I also think that this should be understood as an inherent limitation of these metrics. In the end, however, I don’t expect that the recommended alignment method will be widely adopted for FIs. I expect that, with care, FIs will continue to have valid uses in informing clinical practice and in clinical trials.

We partially agree with this comment. On the one hand, as noted above, we do not expect aligned scores to be adopted for FIs in clinical settings. Indeed, they are not only hard to explain, but also hard to implement in field work. We do believe, however, that the alignment method can get us closer to develop FIs that can prove relevant in clinical settings. 

Specific comments:

8. Lines 1-6: Very well put.

Thank you for noticing it. We putted a lot of work on it to make it as clear as possible.

9. Line 80: The expectation that components should be correlated – I’m not sure I follow this (and what seems to be a counter example, cancer, is discussed later). Might relationships with frailty not be orthogonal?

If FI components were orthogonal, it would not make any sense the use of the alignment method, and our conclusions would not follow. It is our argument, however, that components should be correlated if only because, as noted before in Comment #3, they correlate with biological and chronological aging; that is, they have a common source of variation (i.e., frailty). This alone would induce a measure of non-orthogonality. In the end, it all depends on the agreed measurement model. All our results are predicated on a measurement model that assume deficits are non-orthogonal. Needless to say, there is no logical need for this to be the case, but a case needs to be made if comparisons are to be credible. 

10. Line 84: “max out” – certainly such components do not help with future rates of accumulation, but if comparisons are with less frail subgroups, are these still not useful?

This is a very acute observation, thank you for bringing this to our attention. Indeed, even components that max out early in old age can be useful to score less frail groups, as long as not all components exhibit this behavior. Ideally, we would like to include components with different “severity” to cover as much “frailty terrain” as possible. We are clearer on this matter in the new version. 

11. Line 106: “minimize” – perhaps a different word choice, in that it is not clear that minimization has occurred.

Thank you for bringing this to our attention. This is true, of course. The new version takes care of this imprecision. 

12. Line 118: “self-reported” – a limitation, correct? Is this likely to contribute to some non-invariance?

This is indeed a relevant empirical question that should be pursued in the future. Ideally, data coming from different data collection methods is necessary to explore and quantify the effect of this potential source of invariance. We agree that it certainly can be the case, but there are no a priori reason to think this is the case. 

13. Section 2.3: It is not clear to me why the amount of “explained variability” should be equal for a given component across subgroups. Isn’t it to be expected that the prevalence of components should vary depending on biological age so that some relationships may be more attenuated or vary depending on one’s underlying health status?

Thank you for bringing this lack of clarity from our part to our notice. It is indeed expected that the prevalence of components should vary depending on biological age, along with the variability of the frailty score. The relationships -- the amount of the frailty score’s variability explained by each component--, however, should be roughly the same. The underling measurement model assumes that components increase their likelihood of appearance as frailty increase, irrespective of the population subgroup to which individuals belong. The intuition that the probability of a component’s expression is small for lower levels of frailty is exactly right, the point is that for a given level of frailty, said probability should stay the same irrespective of the subgroup to which the individuals belong, explaining the same amount of variability (across groups) at every frailty level. We are clearer on this point in the new version.

---

## [Decision Letter · Decision Letter 1]

14 Sep 2023

On the comparability of frailty scores under the accumulation of deficits approach

PONE-D-23-10946R1

Dear Dr. Huffman,

We’re pleased to inform you that your manuscript has been judged scientifically suitable for publication and will be formally accepted for publication once it meets all outstanding technical requirements.

Kind regards,

Guillermo Salinas-Escudero, PhD. MsC.

Academic Editor

PLOS ONE

Reviewers' comments:

Reviewer's Responses to Questions

**Comments to the Author**

1. If the authors have adequately addressed your comments raised in a previous round of review and you feel that this manuscript is now acceptable for publication, you may indicate that here to bypass the “Comments to the Author” section, enter your conflict of interest statement in the “Confidential to Editor” section, and submit your "Accept" recommendation.

Reviewer #1: All comments have been addressed

Reviewer #2: All comments have been addressed

2. Is the manuscript technically sound, and do the data support the conclusions?

Reviewer #1: Yes

Reviewer #2: Yes

3. Has the statistical analysis been performed appropriately and rigorously? 

Reviewer #1: Yes

Reviewer #2: Yes

4. Have the authors made all data underlying the findings in their manuscript fully available?

Reviewer #1: Yes

Reviewer #2: Yes

5. Is the manuscript presented in an intelligible fashion and written in standard English?

Reviewer #1: Yes

Reviewer #2: Yes

6. Review Comments to the Author

Reviewer #1: (No Response)

Reviewer #2: My concerns from the initial review have all been adequately addressed. I feel the manuscript is well-written and should be regularly cited.

7. PLOS authors have the option to publish the peer review history of their article (what does this mean?). If published, this will include your full peer review and any attached files.

Reviewer #1: No

Reviewer #2: No

---

## [Editor Report · Acceptance letter]

18 Sep 2023

PONE-D-23-10946R1 

On the comparability of frailty scores under the accumulation of deficits approach 

Dear Dr. Huffman:

I'm pleased to inform you that your manuscript has been deemed suitable for publication in PLOS ONE. Congratulations! Your manuscript is now with our production department. 

Kind regards, 

on behalf of

Dr. Guillermo Salinas-Escudero 

Academic Editor

PLOS ONE